



# Limited effect of organic matter addition on stabilised organic carbon in four tropical arable soils

Marijn Van de Broek[1], Fiona Stewart-Smith[1], Moritz Laub[1], Marc Corbeels[2,3], Monicah Wanjiku Mucheru-Muna[4], Daniel Mugendi[5], Wycliffe Waswa[2], Bernard Vanlauwe[2], and Johan Six[1]

[1]Department of Environmental Systems Science, ETH Zurich, Zurich, Switzerland
[2]International Institute of Tropical Agriculture (IITA), c/o ICIPE Compound, P.O. Box 30772-00100, Nairobi, Kenya
[3]CIRAD, Avenue d'Agropolis, 34398 Montpellier, France
[4]Department of Environmental Sciences and Education, Kenyatta University, P.O. Box 43844-00100, Nairobi, Kenya
[5]Department of Land and Water Management, University of Embu, P.O. Box 6-60100, Embu, Kenya

**Correspondence:** Marijn Van de Broek (Marijn.VandeBroek@usys.ethz.ch)

**Abstract.**

Arable soils are generally characterised by a low soil organic carbon (SOC) content, with negative consequences for soil health, crop yield and global climate. Thus, over the past decades, there has been a focus on how agricultural management practices, such as organic resource addition, can increase the amount of SOC. To sustainably increase SOC stocks, a portion of the organic amendments added to the soil has to be stabilised in persistent fractions such as mineral-associated organic carbon (MAOC). However, there is a lack of research on the magnitude of changes in MAOC in tropical agroecosystems in response to organic resource additions. Here, we show for four long-term field trials in Kenya that the addition of large amounts of organic amendments (farmyard manure or *Tithonia diversifolia* biomass at 4 t C ha$^{-1}$ yr$^{-1}$ for 16 to 19 years) to maize monocropping systems had variable effects on topsoil MAOC stocks (0–15 cm depth), and no significant effect on subsoil MAOC stocks (15–50 cm depth) compared to a control treatment. The addition of mineral N fertiliser did not affect MAOC stocks at any site. Using stable carbon isotopes ($\delta^{13}$C), we found that the portion of topsoil MAOC originating from *Tithonia* biomass was larger in the sandy (25 - 40 %) compared to the clayey soils (0.5 - 12 %), while the portion of total added *Tithonia* biomass that was stabilised over a time period of 16 - 19 years was below 7 % across all sites, or a SOC stabilisation rate of 0.8–27 g C m$^{-2}$ yr$^{-1}$. Using these results, we conclude that while in sandy soils the stabilisation of added OC contributed substantially to limiting SOC losses upon cultivation, this was not the case for clayey soils. These differences were due to the much lower SOC stocks in the sandy soils, compared to the clayey soils. Our results underline the challenges associated with improving soil health in sub-Saharan Africa and stress the need for more research to reliably assess if and how additional SOC can be stabilised over decadal time scales in highly weathered tropical soils.





## 1 Introduction

Soils store a large amount of carbon in the form of organic molecules (Scharlemann et al., 2014). This soil organic carbon (SOC), as part of total soil organic matter (SOM), is a necessary component of soils to provide key ecosystem services such as enabling primary production and climate regulation (Schulte et al., 2014; Oldfield et al., 2019; Van de Broek et al., 2019). However, anthropogenic land use changes, particularly the conversion of natural ecosystems to agriculture, have led to sub-

stantial losses of SOC in the topsoil (Don et al., 2011; Poeplau et al., 2011) and subsoil (Müller et al., 2024; Sanderman et al., 2017). This has driven research efforts aimed at reversing these losses and increasing SOC storage in arable soils (Minasny et al., 2017; Paustian et al., 2016; Chenu et al., 2019; Batjes, 2019; Padarian et al., 2022). However, the feasibility of achieving substantial and long-term SOC sequestration remains the subject of ongoing scientific debate (Schlesinger and Amundson, 2019; van Groenigen et al., 2017; de Vries, 2018; VandenBygaart, 2018; Poulton et al., 2018).

Much of the knowledge related to the effect of land use changes and land management on SOC storage comes from research conducted in economically-developed countries, especially in temperate regions (Beillouin et al., 2022). These studies have shown that multiple agricultural management practices, such as the regular application of organic amendments (Gross and Glaser, 2021; Beillouin et al., 2023), the incorporation of crop harvest residues (Lehtinen et al., 2014) or the cultivation of cover crops (Poeplau and Don, 2015), have the potential to increase, or at least maintain, the amount of topsoil SOC.

In contrast, much less is known about the effects of agricultural management practices on SOC stocks and greenhouse gas emissions in sub-Saharan Africa (SSA) (Rosenstock and Wilkes, 2021). For example, of the 346 long-term agricultural field experiments registered in the GLTEN database (https://glten.org/), only 25 (ca. 7%) are located in SSA. This is problematic, as food demand in SSA is expected to increase significantly in the coming decades. This will require the knowledge to increase crop productivity, while maintaining soil fertility, if SSA is not to rely on already substantial imports of cereals (van Ittersum

et al., 2016). Another challenge is that findings from temperate ecosystems do not necessarily translate to tropical regions, given the differences in soil characteristics and climate. For example, single management practices that increase topsoil OC in temperate agroecosystems (such as no-tillage, crop residue retention and crop rotations; Beillouin et al. (2023)) may not have the same effect in highly-weathered tropical soils, where only the combination of practices, applied consistently over time, have shown to effectively increase SOC (Corbeels et al., 2019). In addition, sustaining soil nutrients and crop yields in SSA

may require a greater reliance on mineral fertilizers compared to temperate agroecosystems (Vanlauwe et al., 2014; Falconnier et al., 2023).

The combined application of organic amendments and mineral fertilizers to maximize the agronomic use efficiency of nutrient inputs and improve crop productivity is part of the Integrated Soil Fertility Management (ISFM) paradigm (Vanlauwe et al., 2010). This has been studied in different regions of SSA (e.g., Rusinamhodzi et al., 2013; Sommer et al., 2018; Sprunger et al.,

2019; Adams et al., 2020; Cardinael et al., 2022), with results showing that the combined application of organic amendments and mineral fertilizer leads to higher maize yields than using either input alone (Chivenge et al., 2011; Gram et al., 2020; Laub et al., 2023b). However, the effect on SOC differs. While in temperate agroecosystems the addition of organic amendments has the potential to maintain or increase SOC stocks (e.g., Gross and Glaser, 2021; Beillouin et al., 2023), many experiments





studying ISFM in SSA report a consistent decrease in SOC stocks over time (e.g., Cardinael et al., 2022; Sommer et al., 2018;

Laub et al., 2023a). Yet, SOC stocks in treatments receiving organic amendments are generally higher compared to a control treatment without organic inputs (Gram et al., 2020). Even at high application rates of organic amendments (4 t C ha$^{-1}$ yr$^{-1}$), SOC stocks have been found to be at best maintained over time, and only at sites with initially low SOC levels (Laub et al., 2023a). In contrast to the results for crop productivity, there is generally no positive interaction between the addition of mineral fertilizer and organic amendments to maintain topsoil SOC stocks (Chivenge et al., 2011; Gram et al., 2020).

While there has been substantial research on the effect of agricultural nutrient management practices on SOC stocks in SSA, as shown by recent meta-analyses (e.g., Chivenge et al., 2011; Powlson et al., 2016; Corbeels et al., 2019; Gram et al., 2020), there is a lack of knowledge regarding the long-term stability of newly-formed SOC. Specifically, two key research questions remain unresolved concerning the potential to store SOC in agricultural soils in SSA: (1) to what extent is SOC stabilized through mineral associations, and (2) whether nutrient management practices influence SOC stocks below the plough layer.

Concerning the first research question, a key method to study SOC stabilization involves separating chemically complex SOC into simplified soil fractions (Heckman et al., 2022). A widely adopted fractionation scheme since its inception by Cambardella and Elliott (1992) separates total SOC into particulate organic carbon (POC) and mineral-associated organic carbon (MAOC), either by size or density fractionation (Cotrufo et al., 2019; Lavallee et al., 2020). Large-scale application of this method has shown that (1) the abundance of POC is controlled by factors favouring SOC loss, while the abundance of MAOC is controlled

by C inputs and the stabilization potential of soil minerals (Hansen et al., 2024), (2) the abundance of POC and MAOC, their ratios, and their sensitivity to climate change differ across land use types (Lugato et al., 2021; Sokol et al., 2022; Hansen et al., 2024) and plant mycorrhizal associations (Cotrufo et al., 2019) and (3) there is a maximum amount of MAOC that can be stabilized in soils (Stewart et al., 2007; Six et al., 2024), which depends on soil mineralogy (Georgiou et al., 2022). However, the data used in the meta-analyses of the aforementioned studies show a notable under-representation of studies from SSA

(Georgiou et al., 2022; Hansen et al., 2024), raising uncertainties about whether the same factors control POC and MAOC abundance in highly weathered tropical soils. Moreover, there is a large diversity in soil types across SSA and different factors that govern the amount of total SOC (von Fromm et al., 2021). As arable soils globally have the largest deficit in stabilized SOC (Georgiou et al., 2022), understanding how management practices in agroecosystems on highly weathered tropical soils affect the amount of stabilized SOC is essential to assess the potential to partially restore, or at least maintain, these stocks

using appropriate management practices.

Concerning the second research question, studies on the effect of agricultural management practices on SOC stocks below the plough layer remain limited compared to studies focused on the topsoil (Yost and Hartemink, 2020). Most studies on the effects of agricultural management practices on SOC in SSA are limited to the top 20–30 cm of soil. The few exceptions (Laub et al., 2023a; Shumba et al., 2024) show that while significant differences in the topsoil SOC stocks were observed,

management practices did not affect SOC stocks below the plough layer. While this lack of detectable differences may be partly explained by the low number of replicates, the findings are not surprising as SOC below the plough layer in tropical (and other) soils is generally very old (Balesdent et al., 2018; von Fromm et al., 2024b; Sierra et al., 2024), indicating that only a small fraction of recently added OC is stabilized below the plough layer (Sierra et al., 2024). As global initiatives to increase





SOC stocks, such as 4 per mille (https://4p1000.org), rely on increasing SOC stocks down to 1 m depth (Minasny et al., 2017),
more research is needed to assess whether or not this is feasible (VandenBygaart, 2018).

In this study, we address the above-mentioned research questions concerning the effect of agricultural nutrient management on stabilized SOC in both the top- and subsoil in SSA. To do so, we performed a density fractionation on soil samples collected from four long-term agricultural field trials in central and western Kenya, where different ISFM practices were applied for 16 and 19 years. We measured SOC stocks of two soil fractions (POC and MAOC) and used stable ($\delta^{13}$C) and radioisotopes
($\Delta^{14}$C) of SOC along the soil profile to quantify the stabilisation of recently added OC through organic amendments. We address the following specific research questions. (1) How do different nutrient management strategies affect stabilised OC stocks (MAOC) in the 0–15 cm depth layer? (2) How do different nutrient management strategies affect POC stocks in the 0–15 cm depth layer? And (3) how do different nutrient management strategies affect MAOC stocks in the 15–50 cm depth layers? The following hypotheses were formulated, partly based on previous research we conducted at the study sites, each
referring to the respective research question. (1) Continuous organic matter addition increases SOC stocks compared to the control treatment by enhancing MAOC stabilization, leading to higher absolute increases in MAOC stocks in clayey soils compared to sandy soils, given the higher content of clay minerals in the former. (2) Continuous organic matter addition leads to higher POC stocks compared to the control treatments. And (3) continuous organic matter addition at the soil surface has no long-term effect on MAOC accumulation in soil layers below 15 cm depth (the plough layer). By addressing these research
questions, this study aims to improve our knowledge on the extent to which different nutrient management practices affect SOC stocks, contributing to climate change mitigation and improving soil health in tropical soils.

## 2   Materials and Methods

### 2.1   Study sites and sample collection

This study used soil samples collected from four long-term agricultural field trials located in central (Embu and Machanga)
and western (Sidada and Aludeka) Kenya, which were established in 2002 and 2005, respectively (Fig. S1). In all trials, a monoculture of maize (*Zea mays*) was grown during two growing seasons per year, coinciding with the long (March - August/September) and short (September/October - January/February) rainy seasons. With the exception of Embu (20.1 °C) , the sites are characterized by similar mean annual temperatures (22.6 - 24.4 °C). The sites in western Kenya receive a larger amount of rainfall compared to the sites in central Kenya (Table 1). In each region, a location with a sandy and clayey soil were
selected. The sites at Embu (Humic Nitisol, 71 % clay) and Sidada (Humic Ferralsol, 57 % clay) are characterised by soils with a clayey texture, while Machanga (Ferric Alisol, 7 % clay) and Aludeka (Haplic Acrisol, 8 % clay) have sandy soils. It it noted that the sandy site at Machanga experienced intense topsoil erosion throughout the experiment.

The field trials were set up to study the effect of different nutrient management strategies, following ISFM principles, on maize yield and SOM dynamics. For all treatments, aboveground crop residues were removed after harvest. For the present
study, four treatments with a varying effect of nutrient management on the topsoil SOC content were selected (based on Laub et al. (2023a)): (1) the control treatment receiving OC only from roots ("Control"), (2) the treatment receiving 120 kg ha$^{-1}$



**Table 1.** General characteristics of the soils and climate at the study sites. The reported soil texture is the average down to 0.5 m depth, weighted by the layer thickness (depth profiles are shown in Figure S2). The initial topsoil OC content was reported by Laub et al. (2023a).

| Site | Region | Clay (%) | Silt (%) | Sand (%) | Initial topsoil OC (%) | MAP (mm yr$^{-1}$) | MAT (°C) |
|---|---|---|---|---|---|---|---|
| Sidada | Western | 56.7 | 43.3 | 0 | 1.5 | 1730 | 22.6 |
| Aludeka | Western | 8.3 | 35.4 | 56.3 | 0.8 | 1660 | 24.4 |
| Embu | Central | 71.3 | 28.7 | 0 | 2.9 | 1175 | 20.1 |
| Machanga | Central | 7.2 | 25.4 | 67.4 | 0.3 | 795 | 23.7 |

mineral N fertilizer (as CaNH$_4$NO$_3$) per growing season (i.e., 240 kg ha$^{-1}$ yr$^{-1}$) ("Control+N"), (3) the treatment receiving 4 t C ha$^{-1}$ yr$^{-1}$ as *Tithonia diversifolia* ("TD") and (4) the treatment receiving 4 t C ha$^{-1}$ yr$^{-1}$ as farmyard manure ("FYM") (Table 2).
While treatments with lower amounts of added organic amendments were present (at 1.2 t C ha$^{-1}$ yr$^{-1}$), the treatments receiving
the largest amount of organic amendments (4 t C ha$^{-1}$ yr$^{-1}$) were chosen as they were shown to having the largest impact on topsoil SOC stocks (Laub et al., 2023a). We chose organic resource treatments that did not receive mineral N, as the addition of mineral N had a negligible effect on the topsoil SOC content in these field trials (Laub et al., 2023a), in line with findings from other sites in SSA (Gram et al., 2020). All treatments, replicated three times in a split-plot design, received a blanket application of 60 kg P ha$^{-1}$ (as triple superphosphate) and 60 kg K ha$^{-1}$ (as muriate or potash) each growing season at planting.
The organic amendments were incorporated into the soil down to a depth of 15 cm using a hand hoe. For a full description of the field trials, we refer to Laub et al. (2023a) and Laub et al. (2023b).

**Table 2.** Overview of the treatments applied at the long-term field trials used in the present study.

| Treatment | Organic amendment | Mineral N |
|---|---|---|
| Control | - | - |
| Control+N | - | 240 kg N ha$^{-1}$ yr$^{-1}$ |
| TD | *Tithonia diversifolia* (4 t C ha$^{-1}$ yr$^{-1}$) | - |
| FYM | Farmyard manure (4 t C ha$^{-1}$ yr$^{-1}$) | - |

Soil samples were collected in 2021 from each of the three replicates of each treatment, at three depth intervals: 0–15 cm, 15–30 cm and 30–50 cm depth (Laub et al., 2023a). In the present study, the 0–15 cm depth layer is referred to as topsoil or plough layer, while the 15–50 cm depth layer is referred to as subsoil. Samples were collected using a half open gauge auger
(60 mm diameter, Eikelkamp Soil & Water) at Embu, or a 55 mm diameter Giddings corer (Giddings Machine Company) at the other sites. Upon sample collection, the soils were sieved at 8 mm, air-dried and stored for further analysis.

A detailed analysis of the effects of all treatments on total topsoil SOC and maize yield has been presented in Laub et al. (2023a) and Laub et al. (2023b), respectively. For the treatments studied here, both maize yield (Fig. S3) and total aboveground biomass productivity (Fig. S4) were significantly higher in the treatments receiving organic amendments (TD or FYM) com-





pared to the control treatment at all sites. A similar trend was observed for the treatments receiving only mineral N at all sites-except Embu. All treatments (with the exception of TD and FYM at 4 t C ha$^{-1}$ yr$^{-1}$ at Aludeka) experienced a decrease in topsoil SOC concentration (0–15 cm) over the duration of the experiment (Fig. S5), with the lowest losses observed in the 4 t C ha$^{-1}$ yr$^{-1}$ FYM treatment, intermediate losses in the 4 t C ha$^{-1}$ yr$^{-1}$ TD treatment, and the highest losses in the control treatment, whether or not mineral N was applied (Laub et al., 2023a). In addition, based on samples collected in 2021, no significant effect of any treatment on the total SOC stock in the 15–50 cm depth layer was observed at any site (Laub et al., 2023a).

## 2.2 Basic soil properties

Soil texture was determined for the control treatment at each site for all three replicates and soil depths (0–15, 15–30 and 30–50 cm depth). To do so, 0.5 g of soil was weighed in glass beakers and placed in a hot water bath (45 °C), and 10 mL H$_2$O$_2$ (35 %) was added five times to decompose all organic matter. After oven-drying, the samples were transferred to plastic tubes and shaken overnight in 7 mL (NaPO$_3$)$_6$ (10 %) to ensure full dispersion of the soil particles. Subsequently, the samples were analysed with a laser diffraction particle size analyser (LS 12 320, Beckman Coulter). Soil particle size classes were defined as clay (< 2 $\mu$m), silt (2 – 53 $\mu$m) and sand (>53 $\mu$m).

Soil pH$_{H_2O}$ was determined by dissolving 10 g of soil in 25 mL milli-Q® water and shaking the mixture for 2 hours on a mechanical shaker. The sediment was allowed to settle for 24 hours, after which the solution was centrifuged for 5 minutes at 4,700 rpm, and the pH of the supernatant was measured.

Soil bulk density was determined by Laub et al. (2023a). The stone-free weight of the dried soil was divided by the volume of the auger used to collect the soil sample, which was corrected for the volume of stones (assuming a stone density of 2.65 g cm$^{-3}$). Outliers in bulk density were identified and removed as follows for the present study. First, it was assessed whether the range in bulk density values for the replicates of the same treatment at the same site and depth interval was larger than 0.25 g cm$^{-3}$ (to assess consistency between the replicates). If this was the case, data points that differed by more than 0.25 g cm$^{-3}$ from the average bulk density for all treatments at the same depth at the same site were identified as outliers (Fig. S7). To determine if the bulk density was significantly different between treatments at the same site, a linear mixed effects model was used (see Section S1.2). As the significant differences were limited to 4 out of the 72 unique site - treatment - depth combinations, it was chosen to use the average bulk density per soil depth at each site to calculate SOC stocks, to avoid non-significant differences in bulk density to influence the results (Fig. S8). To account for the variation in bulk density when calculating stocks of POC and MAOC, the standard deviations of all samples at the same depth and site were used.

## 2.3 Density fractionation

To determine the most suitable fractionation method, a subset of 16 samples from all sites and two soil depths was fractionated into particulate organic matter (POM) and mineral-associated organic matter (MAOM) using a size cut-off of 53 $\mu$m. Note that in this study the terms POM and MAOM are use to refer to the fractions, while POC and MAOC are used to refer to their C content. Both size fractions were subsequently separated by density fractionation (see below) to assess if POM and MAOM were sufficiently isolated, using the OC content of these fractions and a visual inspection under a stereoscope. As





this separation by size appeared not to sufficiently separate both fractions, a density fractionation was used to fractionate all samples.

A total of 144 soil samples was fractionated (4 sites x 4 treatments x 3 depths x 3 replicates). The density fractionation was performed by weighing approximately 5 g of 2 mm-sieved soil into 30 mL of sodium polytungstate (SPT, Sometu, Germany) in a Falcon tube. To determine the optimal density of the SPT, a range of densities between 1.4 and 1.8 g cm$^{-3}$ was used on a subset of samples from all four sites (Cerli et al., 2012). The optimal density of SPT was determined as the one leading to the highest OC content of the light fraction (indicating a better separation of POM from MAOM). This resulted in an optimal SPT density of 1.6 g cm$^{-3}$ for Aludeka, and 1.7 g cm$^{-3}$ for Embu, Machanga and Sidada. Next, the samples were dispersed (in SPT) using ultrasonication at 400 J mL$^{-1}$ (ca. 220 sec at 200 - 240 W) and centrifuged at 4,700 rpm for two hours, to maximize the separation of POM and MAOM. The POM fraction was separated by decanting the floating fraction over a pre-weighed glass fibre filter (Whatman® GF 6) using a vacuum filtration unit. Any remaining SPT on the filter was removed by rinsing the filter and POM with 400 mL of Milli-Q® water. The filters with POM were oven-dried at 60 °C. The remaining SPT was washed from the MAOM fraction by decanting the SPT and filling the Falcon tube with Milli-Q® water. This mixture was vortexed to mix soil and water, centrifuged at 4,700 rpm for up to two hours, after which the water was decanted. This washing step was repeated until the electrical conductivity of the water was below 50 $\mu$S cm$^{-1}$, requiring on average 4 washing steps. For most samples, a flocculant (1:1 mixture of 0.25M CaCl$_2$ and 0.25M MgCl$_2$) was added to ensure the settling of all mineral particles. This flucculant was only added from the second washing step onwards. Samples to which flocculant was added were rinsed four times, as the EC values were increased by the flocculant, preventing to reach 50 $\mu$S cm$^{-1}$. The rinsed mineral fraction was transferred to aluminium trays, oven-dried at 60 °C, and weighed.

The recovery of total soil mass after the fractionation was on average 98.5 %, with OC recoveries being on average 93 % and > 80 % for all except seven samples. It is noted that the OC recovery was calculated using the OC % of a subsample which was ground after sieving, therefore potentially having a higher OC % as small pieces of roots and POM were present in the sample, compared to the non-ground subsample used for fractionation. For the latter, visible pieces of POM were avoided during subsampling for fractionation, which may at least partly explain the lower OC recovery compared to soil mass recovery. The most likely pathway for OC loss during fractionation was during the rinsing of SPT from MAOM, as not all sediment settled during centrifugation before the first rinse when no flocculant was added. Therefore, the loss of OC was accounted for in all following calculations by increasing the mass of MAOM so that the final recovered mass was equal to the initial mass.

## 2.4 Organic carbon and isotopic analysis

The soil fractions (POM and MAOM) were analysed for OC, total N and $\delta^{13}$C. To do so, a subsample containing 0.25–0.5 mg C was weighed into a tin capsule and analysed on a coupled elemental analyser-isotope ratio mass spectrometer (Thermo Flash HT/EA or CE 1110 coupled to a Delta V Advantage). The presence of carbonates was visually checked on a subset of samples by adding 6M HCl to a subsample, with no bubbles being observed. Therefore, also considering the low soil pH at all sites (< 6.5), the samples were not acidified before OC analysis.





For the analysis of radiocarbon (reported as $\Delta^{14}$C), the whole soil and MAOM for one replicate of each treatment were analysed. A homogenised subsample was weighed into a silver cup and fumigated for 72 hours using 37% HCl (Komada et al., 2008). Afterwards, the samples were neutralized for 24 hours at 60 °C over solid NaOH to remove the acid, and packed in tinfoil. Analysis was performed using a Gas Ion Source (Micadas) at ETH Zürich. Measured $^{14}$C/$^{12}$C ratios were normalized using oxalic acid II (NIST SRM4990C). It is noted that due to technical issues, data on $\Delta^{14}$C could not be analysed for all treatments at some sites. Values of $\Delta^{14}$C were converted to radiocarbon ages, after conversion to F$^{14}$C values, using the *F14CtoC14* function from the *rice* package (Blaauw, 2024) in R.

Stocks of particulate OC (POC) and mineral-associated OC (MAOC) were calculated for each sample by multiplying the OC % of these fractions with the mass of the respective soil layer. As the differences in soil bulk density between treatments at the same site were minimal (see above), no equivalent soil masses were calculated, and SOC stocks are reported down to a common depth.

To estimate the portion of SOC from the organic amendments that has been stabilised as MAOC, an end-member model was applied (after rearranging the terms):

$$C_{or} = \frac{\delta^{13}C_{TR} \cdot C_{TOT} - \delta^{13}C_{CT} \cdot C_{TOT}}{\delta^{13}C_{am} - \delta^{13}C_{CT}} \tag{1}$$

where $C_{or}$ is the portion of MAOC derived from the organic resource (%), $C_{TOT}$ the total portion of MAOC (100 %), $\delta^{13}C_{am}$ is the $\delta^{13}C$ value of the organic amendment (‰), $\delta^{13}C_{CT}$ is the $\delta^{13}C$ value of MAOC in the control treatment (‰), and $\delta^{13}C_{TR}$ is the $\delta^{13}C$ value of MAOC in the treatment receiving the organic amendment (‰). As we had no reliable estimate of the $\delta^{13}C$ of the applied manure, only the portion of C directly originating from *Tithonia diversifolia* of MAOC was estimated, assuming a $\delta^{13}C$ value of *Tithonia diversifolia* of -29 ‰ (Kimetu and Lehmann, 2010). To estimate the portion of all C from *Tithonia diversifolia* over the course of the experiment, the absolute amount of MAOC derived from *Tithonia* was calculated by multiplying the total amount of MAOC by $C_{or}$, and dividing this by the total amount of OC from added *Tithonia* residues over all experimental years.

### 2.5 Statistical analyses

The analysis of significant differences in stocks of POC and MAOC for different depth intervals between the treatments per site (Fig. 2, 3, and S13) was done separately per depth layer, after fitting a linear mixed effects model (using the *lme* function from the *nlme* package in R (Pinheiro et al., 2024)) following the procedures outlined in Zuur et al. (2009), with site and treatment as fixed effects and nested site/block as a random effect. A variance structure (varIdent; Pinheiro et al. (2024)) was added to allow for differences in variance between (1) treatments for the MAOC stocks of the uppermost layer, for (2) treatment and site for POC of the uppermost layers, and for (3) POC and MAOC in the subsoil and for all layers combined. Significant differences ($p < 0.05$) were determined using the *emmeans* function from the *emmeans* package (Lenth, 2024), and letters were assigned using the *cld* function from the *multcomp* package (Hothorn et al., 2008).





The statistical significance of the difference in the $\delta^{13}$C values of POC and MAOC between treatments for the same site and depth (Fig. 4) was tested separately for each SOC fraction and site. Similar to SOC stocks, a linear mixed effects model was fitted using the *lme* function in R (Pinheiro et al., 2024) with treatment and depth as fixed effects, and a random intercept for

block. For the $\delta^{13}$C of POC at Machanga, a variance structure (varIdent) was added to allow for differences in the variance of treatment. For all other sites and fractions, a variance structure (varIdent) was added for treatment and depth. Significant differences were identified using the *emmeans* function (*emmeans* package; Lenth (2024)) and letters were assigned using the *cld* function (*multcomp* package; Hothorn et al. (2008)). Statistical analyses to determine significant differences in the portion of POC and MAOC of total SOC (Fig. S14 and S15) are described in section S1.3.

# 3 Results

## 3.1 Soil pH

The studied soils were acidic, with $pH_{H_2O}$ values between 4.5 and 6.5 (Fig. S6). No consistent effect of treatment on pH was observed across all sites, while at individual sites certain patterns were present. At Sidada, a site with a clayey soil, the addition of mineral N fertilizer (control+N) led to lower pH values throughout the soil profile compared to the other treatments, while

at the other sites this effect was only observed in the topsoil. At other sites, TD (Embu) and FYM (Machanga) treatments had higher pH values compared to the other treatments throughout the soil profile.

## 3.2 Organic carbon concentration and C:N ratio of POM and MAOM

The C:N ratios of POM were consistently higher than those of MAOM, indicating a clear separation between the two fractions (Fig. S9). While the C:N ratios of replicates of MAOM for the same treatment were similar (Fig. S10), a larger variation was

observed between the replicates of the POM fractions, mainly at the sandy sites (Fig. S9). At the clayey sites, the OC content of POM was consistently between ca. 35 and 45 %, while at the sandy sites it varied more widely, ranging from ca. 10 to 40 % (Fig. S11). The latter suggests that some MAOC might have been present in the POM fractions at the sandy sites. In contrast, the OC concentrations of the MAOM fractions were more consistent for replicates of the same treatment, indicating minimal contamination with POM (Fig. S11).

The C:N ratio of MAOM was similar for the different treatments at the same site (Fig. S9). The low C:N ratio (C:N < 12) of the MAOM fractions from the control and control+N treatments suggests a minimal direct incorporation of maize-derived OC in the stabilised MAOC fraction, as the C:N of maize root biomass is typically around 20–40 (Ordóñez et al., 2020; Li et al., 2022). The similar C:N ratio at the clayey sites between MAOM on the one hand, and *Tithonia* (12–15, Partey et al. (2011)) and farmyard manure (9.9–15.4, Laub et al. (2023a)) on the other hand, does not allow to draw similar conclusions about the

origin of MAOM for these treatments. At the sandy sites, however, the low C:N ratio of MAOM (values of ca. 6) suggests a limited direct incorporation without microbial processing of any organic resource into MAOM.





## 3.3 Depth profiles of MAOC and POC concentration and stocks

**Figure 1.** Depth profiles of the cumulative concentration of POC (green) and MAOC (orange), relative to total soil mass, for the different measured depth layers. Dots show the individual measurements. Coloured lines show the average values for that treatment, while the grey line shows the POC or MAOC concentrations for the control-N treatment of the same site and the same depth. The two top rows show results for the clayey sites, the bottom two rows show results for the sandy sites.





The majority of SOC throughout the soil profiles was present in the MAOC fraction (Fig. 1). However, the depth trend of the relative contribution of MAOC to total SOC varied inconsistently among the different sites (Fig. S12). For example, at Embu,

the clayey site in central Kenya, the average relative contribution of MAOC was between 85.4–87.9 % in the top 15 cm, with only a limited increase with depth (88.6–93.4 % MAOC in the 30–50 cm depth layer). In contrast, at Sidada, the clayey site in western Kenya, the contribution of MAOC increased from 80.6–83.5 % in the top 15 cm to between 92.8–96.2 % in the 30–50 cm layer. At both sandy sites, the relative contribution of MAOC to total SOC increased with depth, with values of up to 88.7 % MAOC in the topsoil to as high as 97.1 % in the subsoil.

The effect of treatment on total stock of MAOC down to 50 cm was different between the sites (Fig. S13). For the sites in Western Kenya, no significant differences in the amount of MAOC between treatments down to 50 cm were observed: on average between 6.1 and 7.2 kg MAOC m$^{-2}$ at Sidada and between 3.5 and 4.0 kg MAOC m$^{-2}$ at Aludeka. At Embu, the clayey site in central Kenya, the control+N treatment had significantly lower MAOC stocks (on average 6.1 kg MAOC m$^{-2}$) compared to the other treatments (between on average 8.0 and 8.2 kg MAOC m$^{-2}$) down to 50 cm. In contrast, at Machanga, the sandy

site in central Kenya, the FYM treatment had significantly lower MAOC stocks (on average 1.9 kg MAOC m$^{-2}$) compared to the control treatment (on average 2.5 kg MAOC m$^{-2}$) down to 50 cm. However, the loss of surface soil due to soil erosion complicates the interpretation of these results at Machanga.

### 3.4 Particulate and mineral-associated OC stocks in the topsoil

The addition of 4 t C ha$^{-1}$ yr$^{-1}$ in the form of farmyard manure or *Tithonia diversifolia* over 16 and 19 consecutive years at the

central and western sites, respectively, did not have a uniform positive effect on the stocks of POC or MAOC in the soil layer receiving these amendments (0–15 cm) (Fig. 2). At both sandy sites, no significant differences were found in either POC or MAOC stocks between treatments. While the lack of significance for POC might be partially due to the combination of large variability and a low number of replicates (*n*=3), MAOC results were more consistent between the three replicates (Fig. 2). At Machanga, there was even a tendency for lower MAOC stocks in treatments receiving organic amendments, compared to both

treatments without external OC inputs (Fig. 2g). Again here the interpretation of these results is complicated by the presence of soil erosion at this site.

At the clayey sites, no significant differences in POC stocks were observed between treatments at Embu, while at Sidada the TD treatment had significantly larger POC stocks compared to Control+N. Furthermore, there was a tendency for the FYM treatment to have a higher POC content than the treatments not receiving organic amendments. At Embu, the TD treatment

had larger MAOC stocks compared to the Control+N treatment, while at Sidada the FYM treatment had significantly larger MAOC stocks compared to both treatments not receiving organic amendments. In summary, there was no effect of the addition of large amounts of organic amendments on the amount of stabilised SOC in the topsoil of the sandy soils, and a mixed but limited effect in the clayey soils.



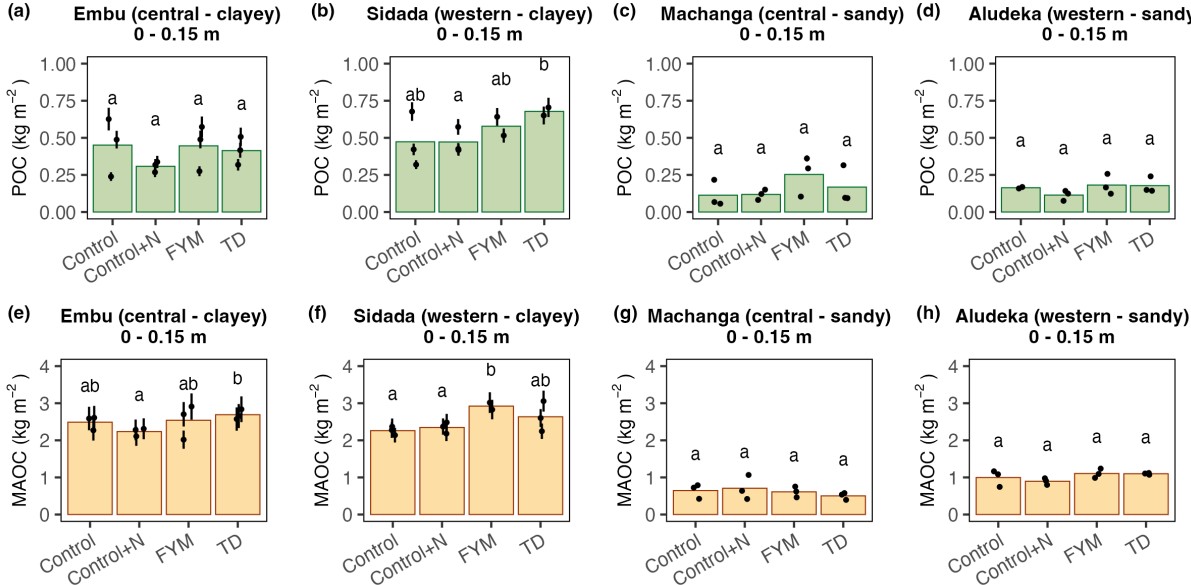

**Figure 2.** Stocks of POC (top row, a-d) and MAOC (bottom row, e-h) for the top 15 cm (expressed as kg C m$^{-2}$ down to 15 cm). Bars show the average stocks, while dots show the individual measurements. Treatments within the same site that do not share any letter are significantly different from each other ($p < 0.05$). The vertical lines show the uncertainty of individual data points due to variability in bulk density ($n = 12$, see section 2.2). These are not visible in case the line was smaller than the dots. No error for the averages (the bars) are shown, due to the low number of replicates ($n=3$).

## 3.5 Particulate and mineral-associated OC stocks in the subsoil

Similarly, in the soil below the plough layer (15–50 cm depth), there were limited differences in POC or MAOC stocks between treatments at the same site (Fig. 3). At the sandy sites, the only significant difference in the POC stocks was between the FYM and TD treatments at Machanga, with neither differing significantly from both treatments not receiving organic amendments. At Aludeka, no significant differences in subsoil MAOC were observed between the treatments, while at Machanga, the FYM treatment had significantly lower MAOC stocks compared to the control treatment.

At the clayey site in western Kenya, Sidada, no significant differences were observed in either subsoil POC or MAOC stocks, while there was a tendency towards lower POC stocks in the treatments receiving organic amendments. At the clayey site of Embu, the Control+N treatment had the lowest POC and MAOC stocks, with POC stocks being significantly lower than those in the FYM and TD treatments, and the MAOC stocks significantly lower than those in the TD treatment. In summary, these results show that at none of the sites, additions of large amounts of organic amendments led to higher MAOC stocks below the

plough layer compared to the control treatments.





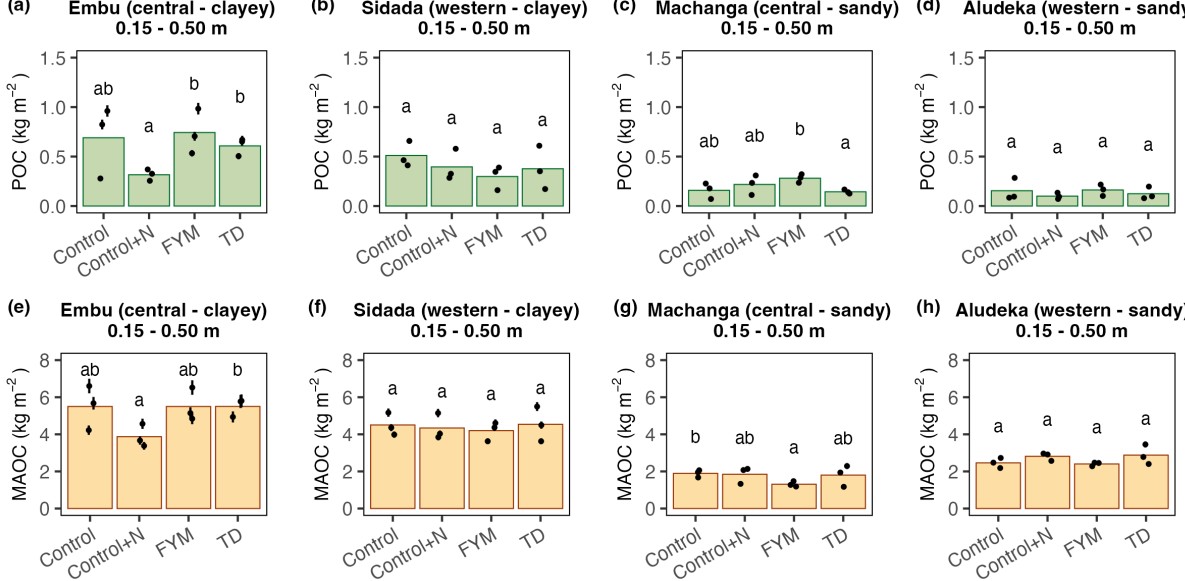

**Figure 3.** Stocks of POC (top row, a–d) and MAOC (bottom row, e–h) below the plough layer (15–50 cm depth; expressed as kg C m$^{-2}$). Bars show the average stocks, while dots show the individual measurements. Treatments within the same site that do not share any letter are significantly different from each other ($p < 0.05$). The vertical lines show the uncertainty of individual data points due to the variation in bulk density between the three replicates. No error for the averages (the bars) are shown due to the low number of replicates ($n$=3).

### 3.6 Sources of stabilized SOC along the soil profile

Stable C isotopes ($\delta^{13}$C) can provide insights into the sources of SOC. Maize crops have a C4 signal ($\delta^{13}$C values of -10 –
-12 ‰, with water and N limitation typically changing the $\delta^{13}$C value of maize up to less than 1 ‰ (Dercon et al., 2006;
Monneveux et al., 2007; Cabrera-Bosquet et al., 2009)). *Tithonia diversifolia* follows the C3 photosynthetic pathway, with a
$\delta^{13}$C value of ca. -29 ‰ (Kimetu and Lehmann, 2010). The $\delta^{13}$C value of the applied manure is unknown, and is likely to have
varied both between the sites and through time, depending on the source of manure.

The variation in $\delta^{13}$C of MAOC in the clayey soils was limited, with no significant differences between any treatments at any
depth (Fig. 4). For the sandy sites, this variation was larger. At Machanga, the $\delta^{13}$C values of MAOC under the FYM treatment
were significantly lower than those in the control treatment throughout the depth profile. In the topsoil, also the TD treatment
had a significantly lower $\delta^{13}$C value compared to the control treatment. At Aludeka, the TD treatment had a significantly lower
$\delta^{13}$C value compared to all other treatments in the topsoil. These results suggest that while at the clayey sites only a limited
portion of MAOC consists of C derived from the organic amendments, at the sandy sites a larger portion of the added C from
farmyard manure and *Tithonia diversifolia* residues was stabilized. The addition of N fertilizer did not have a significant effect
on the stabilisation of added OC (except for the 15–30 cm layer at Machanga), as this would have led to additional maize-
derived OC being stabilised, as this is the sole source of OC in this treatment. It is noted, however, that aboveground biomass



was removed in all treatment, so that only an increase in root biomass upon N fertilization could have contributed to additional OC stabilisation. This implies the absence of a substantial effect of mineral N fertilization on the stabilisation of root-derived OC.

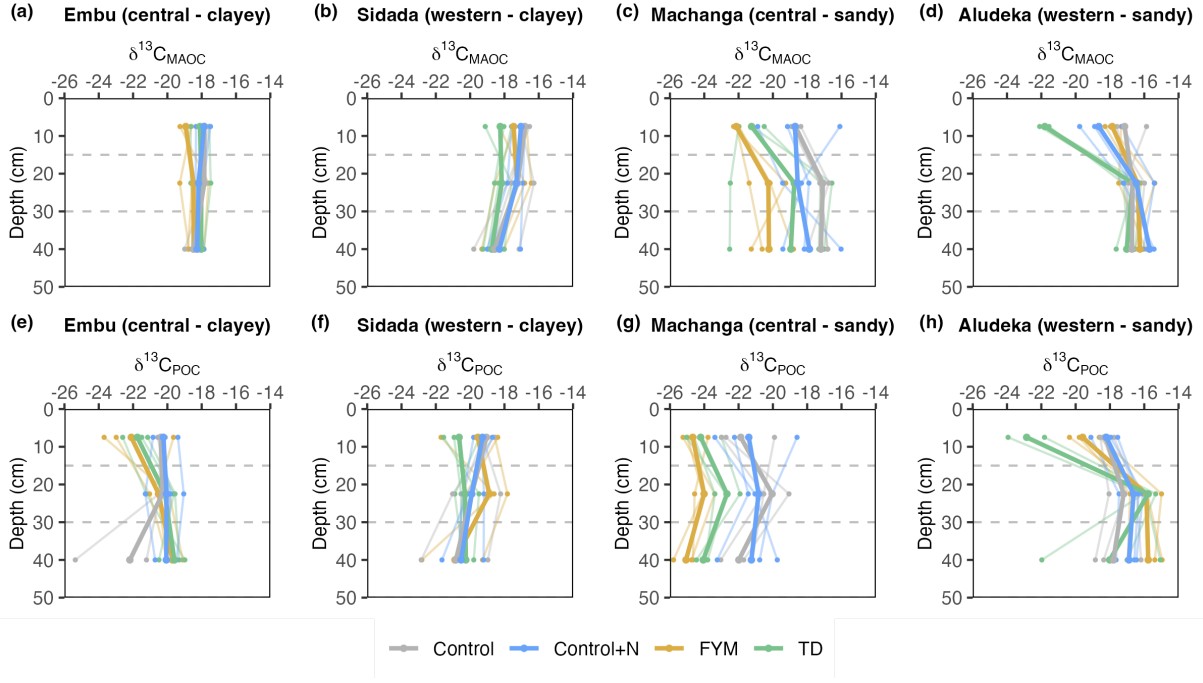

**Figure 4.** Depth profiles of $\delta^{13}$C of MAOC (top row, a-d) and POC (bottom row, e-h) for the different sites. Thin lines show the individual profiles, thick lines show the average per treatment. No letters of significance are shown to maintain readability, but significant differences are presented in the main text.

To estimate the contribution of C from *Tithonia diversifolia* biomass to MAOM in the TD treatment, we used an end-member model with the $\delta^{13}$C values of MAOC for the 0–15 cm depth layer in the control and TD treatments. According to these calculations, the portion of MAOC derived from *Tithonia* is 0.5 % (Embu) and 11.9 % (Sidada) at the clayey sites, and 24.7 % (Machanga) and 39.3 % (Aludeka) at the sandy sites. The overall portion of OC from added *Tithonia* biomass during the entire experimental duration that was stabilised as MAOC (the apparent carbon use efficiency; $CSE_a$) were 0.2 % (Embu) and 4.9 % (Sidada) at the clayey sites, and 1.6 % (Machanga) and 6.7 % (Aludeka) at the sandy sites. When accounting for the absolute amount of MAOC, which was higher at the clayey than the sandy sites, the absolute amount of C from *Tithonia* that was stabilised is on average 313 g C m$^{-2}$ at Sidada, 15 g m$^{-2}$ at Embu, and 124 and 432 g C m$^{-2}$ at Machanga and Aludeka, respectively. This equates to stabilisation rates of 19.6 g C m$^{-2}$ yr$^{-1}$ at Sidada, 0.8 g m$^{-2}$ yr$^{-1}$ at Embu, and 6.5 and 27.0 g C m$^{-2}$ yr$^{-1}$ at Machanga and Aludeka, respectively, at additions of 400 g C m$^{-2}$ yr$^{-1}$.

At the clayey sites, the variation in $\delta^{13}$C for POC between different treatments was larger than for MAOC. At Embu, the only significant difference was found in the topsoil layer, between the TD and control+N treatments, while at Sidada no significant





differences were observed. As for MAOC, there was more variation between the $\delta^{13}$C values of POC at the sandy sites. At Machanga, the FYM treatment resulted in $\delta^{13}$C values that were significantly different from both treatments receiving no organic amendments for all depth layers. The TD treatment had $\delta^{13}$C values that were significantly different from the control+N treatment, but only in the upper two soil layers. At Aludeka, the topsoil $\delta^{13}$C value in the TD treatment was significantly lower

than those in the other treatments, while in the 15–30 cm layer, it was only significantly different from the control treatment. These results show that the amount of organic amendments present in the soil as POC is lower at the clayey sites compared to the sandy sites, where a substantial portion of POC is derived from the added organic amendments, as evidenced by the $\delta^{13}$C values of the TD treatment.

Measurements of $\Delta^{14}$C showed lower values for MAOC compared to the total SOC for all sites and all depths (Fig. 5).

All $\Delta^{14}$C values for MAOC in the topsoil were negative. Topsoil radiocarbon ages of MAOC are between 4 and 551 years at Embu, between 154 and 551 years at Sidada, between 697 and 1397 years at Machanga and between 155 and 407 years at Aludeka. It is noted that because MAOC is not a closed system, these radiocarbon ages are only provided as an indication of the MAOC age. This shows the presence of a substantial portion of old, stabilized SOC. At the sandy sites, the $\Delta^{14}$C values of MAOC in the topsoil of the control treatments not receiving organic amendments were lower compared to the TD and FYM

treatments, showing that more fresh OC is stabilised in the latter. This supports the results that a portion of the added organic amendments was stabilised in the sandy soils, as was shown by the $\delta^{13}$C data. At the clayey Sidada site, differences in $\Delta^{14}$C of topsoil MAOC between the treatments were minimal, in line with the limited differences in $\delta^{13}$C values. Interpretation of the $\Delta^{14}$C data at Embu is complicated by the fact that this site was terraced before the onset of the experiment. Therefore, we refrain from drawing conclusions based on the $\Delta^{14}$C data at this site.

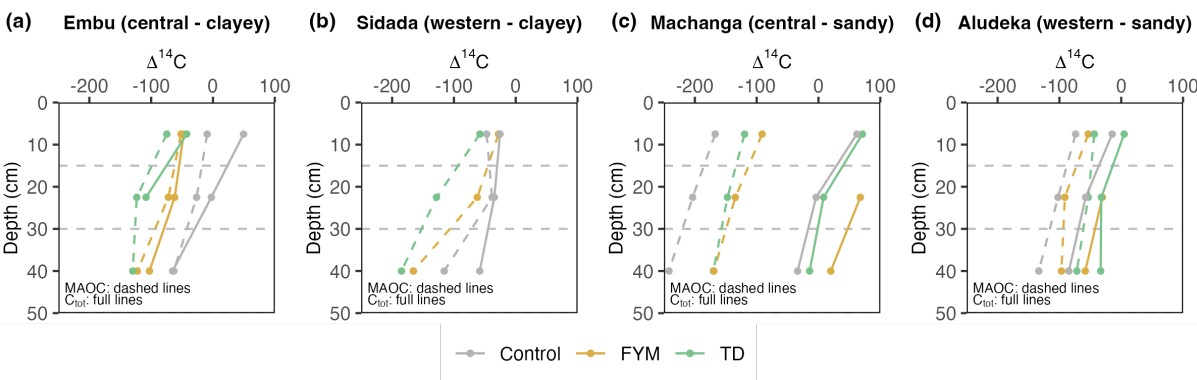

**Figure 5.** Depth profiles of $\Delta^{14}$C values of MAOC (dashed lines) and total SOC (full lines) for the different sites. Note that some data points are missing. As the field experiment at Embu was terraced before the onset of the experiment, this likely affected the observed depth pattern of $\Delta^{14}$C at this site.





## 4 Discussion

### 4.1 Soil organic carbon storage and stabilisation in tropical agroecosystems

The objective of the present study was to evaluate how different organic amendments and mineral N fertilizer affected SOC stabilisation in four tropical field trials of maize monocropping. This extends previous work we presented on how these treatments, among others, affected temporal trends in topsoil OC concentration and SOC stocks (Laub et al., 2023a). Here, we bring together the results from both studies, to summarize these findings and draw comprehensive conclusions about SOC dynamics at these field trials. This is limited to results for the top 15 cm, as no significant differences in subsoil (15–50 cm) SOC stocks (Laub et al., 2023a) and MAOC (this study) were detected, thereby rejecting our third hypothesis. However, it is noted that in another study conducted at Embu, Müller et al. (accepted), a significantly higher SOC stock down to 60 cm was found for the treatment receiving manure, compared to the control treatment. The reason for this difference in results compared to the present study and Laub et al. (2023a) is not clear, and may be related to the higher vertical sampling resolution (5 cm depth intervals) in Müller et al. (accepted). In this section, we use four different sources of information: (1) temporal trends in topsoil SOC concentration over the course of the experiment (Laub et al., 2023a), (2) total SOC stocks for the 0–15 cm depth layer, measured on samples collected after 16 (western Kenya) and 19 years (central Kenya) (Laub et al., 2023a), (3) fractions of POM and MAOM measured on these samples (this study) and (4) measurements of $\delta^{13}$C and $\Delta^{14}$C values of these fractions (this study).

Repeated soil sampling showed that reductions in the topsoil OC concentration were lower in treatments receiving farmyard manure (a loss of 12 %) and *Tithonia* (23 %) at 4 t C ha$^{-1}$ yr$^{-1}$ compared to the control treatment (42 %), across all sites (Laub et al., 2023a). For treatments receiving only mineral N fertilizer at 240 kg N ha$^{-1}$ yr$^{-1}$, declines in SOC concentration were not significantly different from the control treatment. This shows that treatments receiving large amounts of farmyard manure, and to a lower extent *Tithonia*, are the most effective at limiting SOC losses upon cultivation. It should be noted that these data only concern OC concentrations, so no conclusions on quantitative changes in the absolute amount of SOC could be drawn.

Data on topsoil SOC stocks only reflected these results at the clayey sites (Laub et al., 2023a). At Embu, the addition of farmyard manure and *Tithonia* led to higher SOC stocks compared to the control treatment, while this was only the case for farmyard manure at Sidada. At the sandy sites, however, none of the measured SOC stocks of the treatments receiving organic amendments were higher compared to the control, differing from the results using the repeated sampling of OC concentrations. Results for SOC stocks for the treatments receiving only mineral N fertilizer showed no significant difference with the control treatment at any site, aligning with results from temporal trends on topsoil OC concentration. Data from this study show a weaker effect of organic amendments at high rates (4 t C ha$^{-1}$ yr$^{-1}$) on topsoil OC stocks compared to the conclusions derived from repeated sample collection. The lower number of detected significant differences between treatments receiving organic amendments and the control treatment in the current study could be due to the lower number of replicate measurements, which had to be limited to minimize disturbance to the field trials.

We complemented SOC stocks with data on the distribution of topsoil SOC between POC and MAOC (Fig. 1). These showed that MAOC stocks were not consistently significantly higher for treatments receiving organic amendments compared



to the control treatments at the clayey sites, and never at the sandy sites. Therefore, our first hypothesis of MAOC being the

fraction where amended organic C is stabilised is rejected. Also the application of mineral N did not lead to significantly higher MAOC stocks at any of the sites. Furthermore, these data showed the rapid mineralisation of organic amendments in the topsoil. Therefore, also our second hypothesis is rejected. It is noted that these data are subject to the same limitations as the data on total SOC stocks, i.e., a low number of replicates limiting statistical power.

Information on the portion of C from organic amendments that was stabilised in the soil was provided by stable C isotopes.

These data implied a limited stabilisation of added C in the topsoil at the clayey sites. This was confirmed by the calculated values of 0.5 % and 11.9 % of MAOC consisting of *Tithonia*-derived OC at Embu and Sidada, respectively. These translated into apparent carbon storage efficiencies ($CSE_a$) of 0.2 and 4.9 %, respectively, which are in the range of values reported for topical cropland (with an average of 8.2 ± 0.8 %; Fujisaki et al., 2018). These values were lower than the ones calculated using the temporal trends in SOC concentration (11 % and 8 % respectively; Laub et al. (2023a)). At the sandy sites, *Tithonia*-

derived MAOC was 24.7 % and 39.3 % at Machanga and Aludeka, respectively. This resulted in $CSE_a$ values of 1.6 % and 6.7 %, respectively, also lower compared to the values calculated using the temporal trends in OC concentration (2 % and 9 %; Laub et al. (2023a)). The lower $CSE_a$ values calculated using the $\delta^{13}C$ of MAOC compared to calculations based on temporal trends of total SOC were expected. The latter accounts for changes in total SOC (i.e., MAOC and POC), while not being able to differentiate between native SOC and OC originating from organic amendments. This shows the advantage of using $\delta^{13}C$

data in addition to data on OC to quantify actual OC stabilisation.

The $\Delta^{14}C$ data of MAOC added to this that stabilised SOC is likely to be a mixture between relatively young OC, and OC of ages up to thousands of years, given the low $\Delta^{14}C$ values despite a portion of fresh OC inputs being stabilised in the topsoil. These observations are in line with previous studies that found that stabilised SOC is a mixture of organic compounds with ages ranging from years to millennia (Sierra et al., 2024; Schrumpf et al., 2021; Schrumpf and Kaiser, 2015; Baisden et al.,

2002; Koarashi et al., 2012). Furthermore, we observed higher $\Delta^{14}C$ values of MAOC under treatments receiving organic amendments compared to MAOC under the control treatment at the sandy sites, compared to the clayey sites. This confirms the results derived from the $\delta^{13}C$ data, which showed that a larger portion of MAOC at the sandy sites consisted of C originating from organic amendments, compared to the clayey sites.

A synthesis of these results reveals that while treatments receiving farmyard manure and, to a lesser extent, *Tithonia* biomass,

were most effective at limiting reductions in SOC concentration in the topsoil at the clayey sites, this was mainly due to prevented losses of native SOC and only to a limited extent to the stabilisation of added OC. For the sandy sites, in contrast, lower declines in topsoil OC concentration in treatments receiving organic amendments were to a larger extent due to the stabilisation of added OC. The additional information on the distribution of OC in POC and MAOC, and their C isotopes thus adds important information to our understanding of how agricultural nutrient management practices affect the stabilisation of

SOC.





## 4.2 Different response of stabilised SOC to organic amendments in tropical versus temperate ecosystems

Sustainable management of SOC in arable soils is an important prerequisite for maintaining or improving soil health (Lal, 2016; Liptzin et al., 2022), sustaining crop yields (Ma et al., 2023), and preventing further increases in atmospheric $CO_2$ from SOC mineralization (Sanderman et al., 2017; Kopittke et al., 2024). Achieving this requires a clear understanding of how crop
management practices influence MAOC, as the majority of SOC in arable topsoils is associated with soil minerals (ca. 80 % globally; Hansen et al., 2024).

Most studies examining the effect of organic amendments on SOC stabilization in the topsoil have been conducted in temperate and northern agroecosystems. In many long-term experiments, the application of manure or other organic amendments has been shown to lead to significantly higher stabilised SOC stocks (Just et al., 2023; Samson et al., 2020; Herbst et al., 2018;
Trigalet et al., 2014). Although some studies have reported no significant effects (Mayer et al., 2022; Salonen et al., 2023), our results from field trials in Kenya thus contrast with the majority of findings from studies in temperate agroecosystems.

In one study conducted in Ivory Coast, Cardinael et al. (2022) found that while all treatments in a 19-year maize field trial resulted in SOC losses, the treatment receiving compost lost less SOC in the clay and fine silt fraction compared to the treatment without compost. Similar to our results, compost addition mainly led to reduced losses of stabilised SOC, instead of stabilising
freshly-added OC. These findings cast some doubt on the generality of conclusions of meta-analyses suggesting that manure additions increase topsoil SOC in tropical climates (although these conclusions remain uncertain due to limited data from these regions; Gross and Glaser (2021)). They also challenge the assumed potential to increase SOC stocks with conservation agriculture (no-tillage, mulching and crop rotations) in sub-Sahara Africa (Corbeels et al., 2019), as these practices may not necessarily lead to actual stabilisation of additional SOC on soil minerals. Our results therefore suggest that knowledge gained
for temperate agroecosystems cannot be readily applied to tropical ecosystems.

Key reasons for this difference are the distinct mineralogical composition (i.e. low-activity 1:1 clay minerals) and highly weathered nature of tropical soils, which are important controls on the SOC content (Kaiser and Guggenberger, 2003; Rasmussen et al., 2018; Kramer and Chadwick, 2018; Doetterl et al., 2018; Slessarev et al., 2022). These 1:1 clays have less reactive surfaces to stabilise organic molecules, and are known to stabilise less SOC compared to 2:1 clays (Feng et al., 2013;
Georgiou et al., 2022). Specifically for cropland in SSA, it has been shown that the most important controls on the SOC content are oxalate-extractable metals (a measure for secondary poorly crystalline minerals), the chemical index of alteration (a measure for weathering degree) and exchangeable Ca (von Fromm et al., 2021). Given these limitations on the stabilisation of OC in highly weathered soils, once stabilised, organic molecules can remain stabilised by minerals for thousand to tens of thousand of years under natural conditions, as is evident by their high radiocarbon age (von Fromm et al., 2024a). However,
once natural ecosystems are disturbed, stabilised SOC is rapidly lost and it seems not possible to restore the original levels of stabilised SOC within a few decades even by applying large amounts of organic amendments.

Given the limited research on how agricultural nutrient management practices influence SOC stabilisation in tropical agroecosystems, especially in SSA, there is a pressing need for more studies to close this knowledge gap. Expanding this research would not only improve our understanding of the agricultural management practices required to maintain or increase SOC




stocks and the processes governing this, but is necessary to improve SOC models simulating these processes in the tropics. The latter would be an important step forward, as using SOC fractions to improve the initiation, calibration and validation of SOC models can significantly improve the reliability of model outcomes (Herbst et al., 2018; Tougma et al., 2025).

## 5  Conclusions

Our analysis of the fractionation of SOC at four long-term field experiments in Kenya shows that the response to adding
large quantities of organic amendments (farmyard manure or *Tithonia diversifolia* biomass at 4 t C ha$^{-1}$ yr$^{-1}$), with the goal of reducing SOC losses in croplands, differed between clayey and sandy soils. In the studied clayey soils, the portion of C from added organic amendments stabilised on minerals in the topsoil was low, implying that reduced SOC losses in these treatments were mainly due to prevented losses of native SOC. At the sandy sites, in contrast, the lower losses of topsoil OC in treatments receiving organic amendments compared to the control treatment was at least partly due to the stabilisation of OC originating
from the amendments. Below the plough layer (15–50 cm depth), none of the treatments receiving organic amendments had significantly higher MAOC stocks compared to the control treatment. In addition, the application of mineral N fertilizer did not have a significant effect on stocks of stabilised OC at any site. Our results show that rates of SOC stabilisation through organic amendments in arable soils in sub-Sahara Africa may be lower than in temperate agroecosystems, where most research has been conducted to date. More studies in sub-Saharan Africa are therefore needed to more reliably assess the potential
to increase stabilised SOC in croplands, and to identify the most effective management practices for SOC sequestration and consequently sustainable crop production.

*Data availability.*  In the case of acceptance, the data will be published in a publicly accessible repository. For review purposes, the data has been submitted together with the manuscript.



*Author contributions.* The study was designed by MVdB and FSS. The lab work was performed by FSS, and data analysis performed by FSS and MVdB. The long-term field trials are managed by MWMM, DM and WW, and JS and ML were involved in sample collection. The

manuscript was prepared by MVdB, with inputs from all co-authors.

*Acknowledgements.* This research has been supported by the Swiss National Science Foundation (SNSF; Ambizione grant number PZ00P2_193617/1, granted to Marijn Van de Broek). The sampling of the field trials was supported by funds from the European Union's Horizon2020 framework (LANDMARC; Grant agreement ID 869367) and the Swiss National Science Foundation (SNSF; grant number 172940).



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
