# Peer review of "Limited effect of organic matter addition on stabilised organic carbon in four tropical arable soils"

_EGUsphere, 2025_

## Author Comment (AC1)

Feedback from the reviewers is written in italic, while our responses are written in in green. When changes were made to the manuscript, we included a screenshot of this part using track changes.

**Reviewer 1**

*This paper presents a very interesting experiment set in Kenya to explore the impact of agricultural nutrient management on stabilized SOC in a robust experimental design. The introduction strongly stresses the lack of studies on the effects of agricultural management practices on SOC stocks in sub-Saharan Africa.*

*The experiment consists in 4 sites of maize monocropping (1 clayey in central Kenya, 1 sandy in central Kenya, 1 clayey in western Kenya and 1 sandy in western Kenya), with 4 treatments per site (control, control+N, Tithonia diversifolia amendment and farmyard manure amendment), 3 replicates per treatment in each site, and 3 sampling depths, leading to a total of 144 samples. The authors studied SOC, N, δ13C, and Δ14C, and used a size- and density-fractionation protocol to separate the POM and MAOM fractions.*

*The main result is that, unfortunately, long-term, continuous application of OM does not seem to lead to an increase in SOC stocks, neither in topsoil nor subsoil, although it helps slowing down the SOC loss. While the findings themselves form a new and important piece of knowledge, they also highlight the potentially large gap between the results obtained in temperate zones and the field reality in sub-Saharan Africa, stressing the need for more regional studies in the tropical lands. Overall, a very nice and interesting paper!*

We thank the reviewer for taking the time to read our manuscript and for providing detailed and constructive feedback. This is greatly appreciated. Please find our responses to the feedback below.

*Here are some thoughts:*

A summary illustration of the experiment (visually combining Table 1 and Figure S1, and showing the 4 treatments/3 replicates) could be useful. That being said, the experiment is well-described.

Thanks for this suggestion. We believe that the layout of the experiment (4 different treatments at 4 sites, at which 3 depths were samples) is clear from the description in the methods, and from the layout of Fig. 1 in the manuscript, while their location is shown in Fig. S1. Combining all this information in a single figure seems like an overload of information to us. Therefore, we respectfully opted to not include an additional figure to show this layout.

*L180: out of curiosity, do you see an explanation for the different values of SPT needed for good separation?*

Assuming that POM has a similar density across the sites, the optimal density of SPT to separate POM from MAOM depends on the density of the minerals. Below the optimal density, not all POM is floating in SPT, which is detected by a higher OC content of the MAOM compared to the optimal density. Above the optimal density, minerals start to float in the SPT, which is detected by a lower OC content of the POM (Cerli et al., 2012; doi.org/10.1016/j.geoderma.2011.10.009). The lower optimal SPT density of $1.6 \text{ g cm}^{-3}$ at Aludeka therefore indicates that the density of minerals at that site is slightly lower than those at the other sites where the optimal SPT density is $1.7 \text{ g cm}^{-3}$. We note that the optimal SPT density for soils collected in Mediterranean to temperate regions was $1.6 \text{ g cm}^{-3}$ according to Cerli et al. (2012). Using the same methods, our result of an optimal SPT density of $1.7 \text{ g cm}^{-3}$ thus shows the importance of testing the optimal density for soils with different mineralogies, such as tropical soils, to ensure an optimal cut-off between POM and MAOM. It is also worth noting that Cerli et al. (2012) did not test an SPT density of $1.7 \text{ g cm}^{-3}$ (the next higher density was $1.8 \text{ g cm}^{-3}$), so it's not possible to assess whether a density of $1.7 \text{ g cm}^{-3}$ would also have been appropriate for their soils.

In the manuscript, we now make it clear that we tested densities in steps of $0.1 \text{ g cm}^{-3}$, as this information was lacking: "To determine the optimal density of the SPT, a range of densities from 1.4  to $1.8 \text{ g cm}^{-3}$, in $0.1 \text{ g cm}^{-3}$ increments, was used on a subset of samples from all four sites, following the methods outlined by Cerli et al. (2012).".

> in a Falcon tube. To determine the optimal density of the SPT, a range of densities  from 1.4  to 1.8 g cm$^{-3}$, in 0.1 g cm$^{-3}$ increments, was used on a subset of samples from all four sites, following the methods outlined by Cerli et al. (2012). The optimal density of SPT was determined as the one leading to the highest OC content of the light fraction

*L189: how did you make sure to retrieve all your MAOM during the rinsing of the floculant, since you added floculant because you couldn't retrieve all the floating MAOM?*

Thank you for bringing this to our attention. Based on your question, it seems we did not clearly formulate this part of the procedure. We did not intend to rinse out the flocculant. Rather, because the addition of the flocculant affects the electrical conductivity (EC) values, this measure could not be used to determine when sufficient SPT was rinsed out of the samples, as EC values remained high due to the presence of the flocculant. Therefore, using additional samples to which no flocculant was added (which resulted in substantial losses of minerals, so these samples were not used for further analyses), we determined that four rinses were sufficient to reach EC values below $50 \text{ µS cm}^{-1}$.

To avoid any confusion, we now reformulated this in the manuscript to: "As EC values are increased by the presence of the flocculant, preventing to reach values below 50 µS cm$^{-1}$, all samples to which flocculant was added were rinsed four times. This number of rinses was shown to be sufficient for additionally processed samples from all sites, to which no flocculant was added. Rinsing those samples four times resulted in EC values below 50 µS cm$^{-1}$.".

> second washing step onwards.  As EC values are increased by the presence of the flocculant, preventing to reach
> 205  values below 50 $\mu$S cm$^{-1}$, all samples to which flocculant was added were rinsed four times. This number of rinses was shown to be sufficient for additionally processed samples from all sites, to which no flocculant was added. Rinsing those samples four times resulted in EC values below 50 $\mu$S cm$^{-1}$.

*L196: how was the subsampling done, by hand or with an automatic subsampler?*

The subsample was collected by hand. We added this to the manuscript as follows: "For the latter, visible pieces of POM were avoided during the manual subsampling for fractionation, which may at least partly explain the lower OC recovery compared to soil mass recovery.".

> the sample, compared to the non-ground subsample used for fractionation. For the latter, visible pieces of POM were avoided during the manual subsampling for fractionation, which may at least partly explain the lower OC recovery compared to soil mass recovery. The most likely pathway for OC loss during fractionation was during the rinsing of SPT from MAOM, as not

*L197: any other plausible pathway for loss?*

The other potential loss pathway is the loss of both POM and MAOM that stuck to the sonicator tube. As this was minimal, and because we didn't have a reliable way of distributing lost C between POM and MAOM, we assumed that the main loss was through losses of MAOM during rinsing the SPT, as this loss pathway was clearly observed.

*L257: would you say the C:N data (fig.S9) support the idea of 'MAOM-contaminated' POM in sandy soils?*

Yes, the relatively low C:N values of POM in some of the treatments at the sandy sites are in line with the lower OC concentrations of POM for these treatments, compared to the clayey sites. Even though we tested for the optimal SPT density, this suggest that the POM fraction was nonetheless contaminated with minerals.

This is now included in the manuscript in section 3.2 as follows: "The C:N ratios of POM were consistently higher than those of MAOM, indicating a separation between the two fractions, although with a varying degree between sites (Fig. S9)." And: "The latter suggests that some MAOC might have been present in the POM fractions at the sandy sites, which is supported by the lower C:N ratios of POM in certain treatments at the sandy sites (Fig. S9).".

> 275    The C:N ratios of POM were consistently higher than those of MAOM, indicating a  separation between the two fractions, although with a varying degree between sites (Fig. S9). While the C:N ratios of replicates of MAOM for the same treatment

>        varied more widely, ranging from ca. 10 to 40 % (Fig. S11). The latter suggests that some MAOC might have been present in
> 280    the POM fractions at the sandy sites, which is supported by the lower C:N ratios of POM in certain treatments at the sandy sites (Fig. S9). In contrast, the OC concentrations of the MAOM fractions were more consistent for replicates of the same

*L304: you specified the site of Machanga was subjected to strong erosion; is the erosion homogeneous on the whole site (all treatments equally affected)? Did you quantify it?*

The field trial at Machanga has only a very minor slope (estimated to be < 1 %), along which erosion took place (which was initially not expected). The erosion rate has, unfortunately, not been quantified. As there were no visual indications of preferential erosion at certain parts of the field, and the treatments were distributed across the field, there is no reason to assume that certain treatments were affected more by soil erosion than others.

Based on this comment and a comment by the other reviewer, we now describe this in more detail in the methods section where the sites are described (section 2.1): "The sandy site at Machanga had a gentle slope (< 1 %) and therefore experienced topsoil erosion throughout the experiment. Because treatments were randomized within horizontal blocks following the contour of the slope—and erosion affected the field broadly—it is unlikely that any treatment was disproportionately impacted. However, the erosion may have removed part of the topsoil and brought subsoil closer to the surface, causing some of the original subsoil (i.e., below 15 cm depth) to be included in the 0–15 cm samples.".

it noted that the The sandy site at Machanga experienced intense had a gentle slope (< 1 %) and therefore experienced topsoil

erosion throughout the experiment. Because treatments were randomized within horizontal blocks following the contour of the slope—and erosion affected the field broadly—it is unlikely that any treatment was disproportionately impacted. However, the erosion may have removed part of the topsoil and brought subsoil closer to the surface, causing some of the original subsoil (i.e., below 15 cm depth) to be included in the 0–15 cm samples.

125

*L331: if I get it right, the portion of TD-derived MAOC can be quite high. However, you stated above that the MAOC stocks weren't much affected by the treatments. Would this mean that the turnover time of this fraction is (at least for part of it) rather short? It is interesting that these portions of "younger" MAOC are higher in the sites with quite high MAOC Δ14C ages.*

Indeed, the TD-derived MAOC, as a portion of total MAOC was higher at the sandy sites compared to the clayey sites. However, because of the much lower MAOC stocks at the sandy sites, the total amount of stabilized TD-derived MAOC was not consistently higher at the sandy sites.

The input rate of *Tithonia* residues at all sites was equal. Given that turnover time of MAOC = MAOC stock / input rate, and MAOC stocks were substantially smaller at the sandy sites compared to the clayey sites, this suggests that the MAOC turnover rate was indeed faster at the sandy sites. However, as we did not actually calculate the turnover times, we prefer not to mention this in the manuscript.

*L367: wasn't your 3rd hypothesis: 'continuous organic matter addition at the soil surface has no long-term effect on MAOC accumulation in soil layers below 15 cm depth'? I don't see how it is rejected.*

Thanks for noticing this, that is correct (as also pointed out by the other reviewer). This is now corrected in the manuscript as follows: "This is limited to results for the top 15 cm, as no significant differences in subsoil (15--50 cm) SOC stocks (Laub et al., 2023) and MAOC (this study) were detected, thereby confirming our third hypothesis.".

390     dynamics at these field trials. This is limited to results for the top 15 cm, as no significant differences in subsoil (15–50 cm) SOC stocks (Laub et al., 2023a) and MAOC (this study) were detected, thereby  confirming our third hypothesis.

---

## Author Comment (AC2)

Feedback from the reviewers is written in italic, while our responses are written in in green. When changes were made to the manuscript, we included a screenshot of this part using track changes.

**Reviewer 2**

The study described by Van de Broek et al capitalizes on a long term ISFM field experiment in central Kenya (Embu and Machanga) and western Kenya (Sidada and Aludeka) Kenya. The study builds on extensive SOC research performed at the same sites and published in multiple papers by Mortiz Laub et al.

This study adds information to the previous work by measuring MAOM and POM fractions. d13C and D14C radioisotopes on a selection of the ISFM treatments.

This study is a valuable addition to the previous publications as it gained new insights on the fractions of MAOC and POC and the stabilisation of amended carbon in MAOC.

Variable topsoil MAOC stocks in the topsoil as an effect of nutrient or organic amendment application (Tithonia diversifolia or Farm Yard manure addition)

Clear differences in MAOC stocks between in clayey and sandy sites.

No significant differences in SOC fractions in the subsoil (15-50cm).

Previous studies (Laub et al 2023) showed that organic amendments reduced native SOC losses from these field sites. This study showed that the prevented losses in the clayey sites were not caused by the formation of newly formed MAOC from added organic amendments but originated from prevented losses of native SOC while in the sandy soils the lower losses were partly caused by new MAOC stabilisation.

We thank the reviewer for taking the time to read our manuscript and for providing detailed and constructive feedback. This is greatly appreciated. Please find our responses to the feedback below.

More specific comments on the manuscript:

Line 11: Given the distinct stable carbon isotopes signatures of C3 and C4 substrates of Maiz and Tithonia we calculated.. etc.

Thanks for this suggestion. The start of this sentence is now changed to: "Using the distinct stable carbon isotopic signature ($\delta^{13}$C) of the maize crop (C4) and the *Tithonia* amendments (C3), we calculated that [...]".

> N fertiliser did not affect MAOC stocks at any site. Using  the distinct stable carbon isotopic signature ($\delta^{13}$C)  of the maize crop (C4) and the *Tithonia* amendments (C3), we calculated that the portion of topsoil MAOC

Line 96: You now name the treatments nutrient management strategies. However before you this term was not used and only organic amendments and mineral fertilizer addition was used to indicate the different treatments. I would replace "nutrient management strategies" with "amendments".

Thanks for this suggestion, we agree that it's better to stick to one term. "Organic resources" is now used throughout the manuscript, as this term is generally used in research concerning integrated soil fertility management.

Line 98: why a question on the15-50cm layer and measurements in two layers 15-30, 30-50 cm? See also line 134.

In our study, we made use of previously collected soil samples (as described in Laub et al., 2023; https://doi.org/10.5194/soil-9-301-2023). We wanted to get insights in the distribution of C fractions along the depth profile, initially analysing the layers 0-15, 15-30, and 30-50 separately. However, the results were similar for the 15-30 and 30-50 layers, and to simplify the presentations of the results we chose to present them together (i.e., as being the subsoil). At the same time, we want to give the reader the opportunity to see how the depth profiles of C fractions looked like, which is relevant, for example, if one later aims to simulate these depth profiles.

To make this clear to the reader, this sentence is changed to: "In the present study, the 0-15 cm depth layer is referred to as topsoil or plough layer, while the samples collected in the 15-30 and 30-50 cm depth intervals are referred to as the subsoil. Depth profiles of the analysed variables are presented for all layers, while statistical results are presented for the topsoil and the combined subsoil layers.".

> 15–30 cm and 30–50 cm depth (Laub et al., 2023a). In the present study, the 0–15 cm depth layer is referred to as topsoil or plough layer, while the  samples collected in the 15–30 and 30–50 cm depth intervals are referred to as
> 145  the subsoil. Depth profiles of the analysed variables are presented for all layers, while statistical results are presented for the topsoil and the combined subsoil layers. Samples were collected using a half open gauge auger (60 mm diameter,

Why then use alle layers in figure 1 and in text for instance Line 324.

The reason for referring to these layers separately in the text, and showing them separately in the figures, is to provide the reader with this additional level of detail, rather than discussing only the results for both layers combined. As the results for the two subsoil layers were identical, these are used together to formulate the conclusions. We think this is the most straightforward way to discuss while showing the detailed data.

Line 117: For the reader it is difficult to assess what the influence is of the intense topsoil erosion throughout the experiment at the Machanga site. As the site is included in all the analysis the erosion is not strong enough to exclude this site from this study? No bias expected. This could be explained a bit more.

As soil erosion took place across the field, and treatments were distributed in blocks, it is unlikely that erosion differently affected the treatments. However, soil erosion was unfortunately not quantified. Nevertheless, we chose to include the results in our manuscript as they are still useful to the reader. Based on this comment and a comment by the other reviewer, we now describe this in more detail in the methods section where the sites are described (section 2.1) as follows: "The sandy site at Machanga had a gentle slope (< 1 %) and therefore experienced topsoil erosion throughout the experiment. Because treatments were randomized within horizontal blocks following the contour of the slope— and erosion affected the field broadly—it is unlikely that any treatment was disproportionately impacted. However, the erosion may have removed part of the topsoil and brought subsoil closer to the surface, causing some of the original subsoil (i.e., below 15 cm depth) to be included in the 0–15 cm samples.".

>  The sandy site at Machanga  had a gentle slope (< 1 %) and therefore experienced topsoil
>
> **4**
>
> erosion throughout the experiment. Because treatments were randomized within horizontal blocks following the contour of the slope—and erosion affected the field broadly—it is unlikely that any treatment was disproportionately impacted. However, the
> 125 erosion may have removed part of the topsoil and brought subsoil closer to the surface, causing some of the original subsoil (i.e., below 15 cm depth) to be included in the 0–15 cm samples.

Table 1 Could the SOC stocks from Laub et al 2023 be added to this table to make a clear distinction between what is new and what is already published in previous papers?

Table 1 describes general characteristics per site, while the SOC stocks differed per treatment at each site. This results in 16 SOC stocks (4 treatments at 4 sites), which cannot be mentioned in this table without expanding greatly. In addition, Laub et al. (2023; https://doi.org/10.5194/soil-9-301-2023) calculated SOC stocks on an equivalent soil mass basis, while we report stocks of POC and MAOC down to a common depth (0.5 m, as explained in section 2.2). These different ways of reporting mean that both results are not directly comparable. Therefore, we prefer not to mention the SOC stocks by Laub et al. (2023) in the manuscript.

However, to make it clear that these results have been reported by Laub et al. (2023), we now added the following sentence to section 2.1, after we noted that Laub et al. (2023) found no significant differences in the SOC stocks below 15cm depth at any of the sites: "The initial topsoil OC content and SOC stocks across depth, as measured in 2021, were reported by Laub et al. (2023a).".

> (Laub et al., 2023a). The initial topsoil OC content and SOC stocks across depth, as measured in 2021, were reported by (Laub et al., 2023a).

Why was the HCL, NaOH procedure used for the D14C radioisotopes while this was skipped for the OC procedures (Line 203-205)

We did not acidify the samples prior to OC analysis as all samples had a pH < 6.5, and we found no presence of $CaCO_3$ in a subset of samples to which HCl was added (as described in section 2.4). However, the addition of HCl and neutralization with NaOH is standard practice in the lab performing the $^{14}C$ analysis which happens irrespective of the possibility of $CaCO_3$ being present. Therefore, these steps were performed and reported in the manuscript, although it is highly unlikely that it had any effect on the results.

Line 214-215: Not very clearly explained here. What is meant with equivalent soil masses of the respective soil layer, no bulk density of the layers used, Why not? What is meant with a common depth.

Thanks for pointing out that this has not been described properly. We now changed this part to the following and hope to answer your question with this modification: "Stocks of particulate OC (POC) and mineral-associated OC (MAOC) were calculated by multiplying the OC % of each fraction with the mass of the respective soil layer. It has been observed that treatments in field trials can lead to changes in bulk density over time. When this occurs, it is preferable to report SOC stocks down to an equivalent soil mass rather than to

a fixed depth, to avoid bias caused by collecting more (or less) soil in certain treatments at the same sampling depth (Ellert and Bettany, 1995). In our study, however, bulk density differences between treatments within each site were minimal (see above) indicating that the same amount of soil was present in every depth layer at the same site for the different treatments. Therefore, POC and MAOC stocks were reported down to a common depth, rather than for an equivalent soil mass."

> Stocks of particulate OC (POC) and mineral-associated OC (MAOC) were calculated  by multiplying the OC % of  each fraction with the mass of the respective soil layer.  It has been observed that treatments in field trials can lead to changes in bulk density over time. When this occurs, it is preferable to report SOC stocks down to an equivalent soil mass rather than to a fixed depth, to avoid bias caused by collecting more
> 235 (or less) soil in certain treatments at the same sampling depth (Ellert and Bettany, 1995). In our study, however, bulk density differences between treatments within each site were minimal (see above)  indicating that the same amount of soil was present in every depth layer at the same site for the different treatments. Therefore, POC and MAOC stocks were reported down to a common depth, rather than for an equivalent soil mass.

Line 223: Can at least a range of d13C be mentioned for the Farm yard manure?

We understand that this information could be interesting for the reader. However, given that this has not been measured at any time during the experiment, and because the $\delta^{13}C$ depends on what the animals producing the manure were fed (C3, C4, or a combination of both), there is likely a large range in potential $\delta^{13}C$ values of the manure. To not bias the reader with uncertain or potentially incorrect estimates, we prefer not to include speculative values.

Line 367 not correct see also other reviewer

Thanks for pointing this out, this has been corrected in the manuscript.

> 390 dynamics at these field trials. This is limited to results for the top 15 cm, as no significant differences in subsoil (15–50 cm) SOC stocks (Laub et al., 2023a) and MAOC (this study) were detected, thereby  confirming our third hypothesis.

Line 388 not OC stocks but SOC stock?

As we mention "topsoil OC" stocks here, we prefer not the repeat the 'soil' in SOC, and stick to topsoil OC. Otherwise, one would read it as topsoil soil organic carbon.

Line 398: would you expect to find differences with higher numbers of replications? Be more explicit here and a bit less

Thanks for pointing this out. We now added the following sentence: "Although it is uncertain whether more replications would have led to the detection of more significant differences, we encourage future studies to aim for the collection of more replicates, whenever possible.".

> 415 replicate measurements, which had to be limited to minimize disturbance to the field trials. Although it is uncertain whether more replications would have led to the detection of more significant differences, we encourage future studies to aim for the collection of more replicates, whenever possible.

Line 402: Which are in the lower range. 0,2%-4.9% is a lot lower than the average 8,2%!

Thanks for pointing this out. It seems Fujisake et al. (2018; https://doi.org/10.1016/j.agee.2017.12.008) reported standard errors, not standard deviations (they do not clarify this for the value of 8.2 +/- 0.8 % they report, but they note for similar means in their manuscript that the spread is reported as a standard error). This would mean that the mean +/- standard deviation is 8.2 +/- 8.1 % ($n$ = 102). Therefore, our values of 0.2 and 4.6 % are within one standard deviation of their mean. That our values are in the lower range of values they report, but not against the lower limit, also appears from their Fig. 5 (see below).

Therefore, we prefer to keep the statement in our manuscript, but clarify it as follows: "These translated into apparent carbon storage efficiencies (CSE$_a$) of 0.2 and 4.9 %, respectively, which are in the range of values reported for tropical cropland by Fujisaki et al. (2018), with an average of 8.2 ± 8.1 % (mean ± standard deviation).

> calculated values of 0.5 % and 11.9 % of MAOC consisting of *Tithonia*-derived OC at Embu and Sidada, respectively. These translated into apparent carbon storage efficiencies (CSE$_a$) of 0.2 and 4.9 %, respectively, which are in the range of values reported for topical cropland  by Fujisaki et al. (2018), with an average
> 430 of 8.2 ± 8.1 % (mean ± standard deviation). These values were lower than the ones calculated using the temporal trends in

[Figure]

**Fig. 5.** Histogram of the distribution of the conversion rate of C inputs to ΔSOC, i.e. the ratio between ΔSOC and C inputs (n = 102).

*Fig. 5 from Fujisake et al. (2018; https://doi.org/10.1016/j.agee.2017.12.008)*

There is a pressing need for more studies closing the knowledge gap on SOC stabilisation in tropical ecosystems. However, more long-term field trials are needed in tropical ecosystems also in other regions and countries to get representative data.

We fully agree, and hope that future research project can complement our results with more data on SOC stabilization from other tropical regions around the world.